# PROTOTYPICAL ENVIRONMENT-AWARE PROXY WITH COORDINATED OPTIMIZATION FOR MULTI-AGENT DYNAMICS MODELING

## ABSTRACT

Modeling multi-agent dynamic systems is crucial for understanding collective behaviors in various scientific domains. While graph ordinary differential equation (ODE) approaches effectively capture continuous dynamics from irregular data, their performance struggles to generalize across temporal and parameter-induced shifts and degrades severely under potential out-of-distribution fluctuation. In this paper, we propose a novel approach named Prototypical Enviroment-aware Proxy with Coordinated Optimization (PEACE) for multi-agent dynamics modeling. The core of our PEACE is to learn a set of proxy models to simulate environment information while keeping the primary model fixed. In particular, our primary model utilizes temporal graph neural networks to extract invariant observation embeddings across different nodes. More importantly, a range of prototypical prompts are introduced to model temporal distribution shifts with graph ODEs, which are further incorporated with our observation embeddings to serve as proxy models. These proxy models would further generate diverse predictions of unseen trajectories, which are selected by a vision language model for data augmentation. To jointly learn the primary and proxy models, a bi-level strategy is adopted for alternative optimization. In the lower level, we update prompt parameters in the proxy models with our primary model frozen. In the upper level, we integrate all these proxy models and measure the gradient coordination to update our primary models. Extensive experiments on multiple real-world system dynamics datasets demonstrate the superiority of PEACE over state-of-the-art baselines, confirming its effectiveness and robustness.

## 1 INTRODUCTION

Multi-agent dynamic systems are prevalent in real-world settings, ranging from physical simulation (Shao et al., 2022; Xu et al., 2023) to molecular dynamics (Li et al., 2022b; Wu et al., 2023). In such systems, agents interact through behavioral or mechanical influences, resulting in complicated trajectories that could unfold with an evolving graph structure, where nodes represent agents and edges encode their interactions (Kipf et al., 2018). Consequently, modeling and understanding these interacting dynamics are of fundamental importance, as they uncover the governing principles of collective behavior and enable reliable prediction of the system (Cini et al., 2025).

In literature, a variety of deep learning methods have been developed to model multi-agent dynamic systems (Hajiramezanali et al., 2019; Pfaff et al., 2020; Han et al., 2022; Zhang et al., 2024a). Typically, these approaches employ graph neural networks (GNNs) to encode the agent states from their trajectories at the beginning and iteratively aggregate information from neighboring nodes with the message passing mechanism (Kipf & Welling, 2017; Veličković et al., 2018; Xu et al., 2019). These discrete models, which approximate the system dynamics at fixed timestamps, usually suffer from irregularly sampled data and typically require complete observations of every node at every timestamp (Huang et al., 2020; 2021). To mitigate this issue, Ordinary Differential Equations (ODEs) (Chen et al., 2018) are incorporated to capture the system dynamics in a continuous manner. These graphODE efforts (Luo et al., 2023; Wu et al., 2024; Qin et al., 2024; Liu et al., 2025) have demonstrated strong capability in modeling long-range dynamics and effectively learning from irregularly sampled and partially observed data.

Despite their success, most of these approaches are built on the independently and identically distributed (i.i.d.) assumption that training and test data are generated in-distribution (ID) from one single system, which rarely holds in real-world scenarios. In practice, the observed trajectories are typically drawn from multiple systems across diverse environments (Huang et al., 2023; Luo et al., 2024; Wan et al., 2025). And these approaches consequently exhibit degraded and unstable predictions when applied to new environments (Zhang et al., 2024b; Li et al., 2025b). Recently, significant efforts have been devoted to graph OOD generalization, aiming to improve model performance under environment-induced distribution shifts. These approaches include strategies such as data augmentation (Sui et al., 2023; Lu et al., 2024), (causal) invariant representation learning (Li et al., 2022a; Liu et al., 2023) and new model architecture (Yang et al., 2023; Guo et al., 2024).

However, developing such a generalized graph ODE to learn multi-agent system dynamics under distribution shift remains the following two key challenges: ❶ *Coping with diverse distribution shifts.* Dynamical systems often face both temporal distribution shifts (e.g., trajectories drift during long-term temporal evolution) (Zhang et al., 2022b) and parameter-based shifts (e.g., varying temperatures for particle systems or distinct pressure conditions in fluid dynamics) (Baradel et al., 2020; Sanchez-Gonzalez et al., 2020). These shifts perturb the underlying data distribution in fundamentally different ways, making it difficult for a single model to maintain robustness across environments. ❷ *Mitigating mid-train OOD fluctuation.* The training process suffers from instability when the model fails to accumulate and retain knowledge obtained during temporal evolution. As illustrated in Fig 1, the prediction performance of PGODE (Luo et al., 2024) under both ID and OOD settings progressively degrades as the forecasting horizon increases.

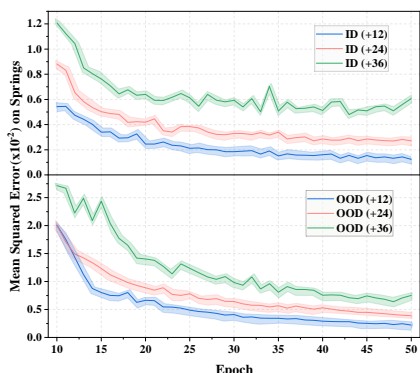

Figure 1: Performance of PGODE on the Spring system under ID and OOD settings with different prediction length.

In particular, features derived from earlier steps tend to be distorted as training progresses, resulting in unstable dynamics learning and degraded generalization.

Towards this end, in this paper, we propose a novel **P**rototypical **E**nvironment-**A**ware Proxy with **C**oordinat**E**d framework (termed **PEACE**) for multi-agent dynamics modeling, which investigates temporally adaptive prototypical prompts as proxy models to simulate environment information and measure the gradient coordination for the primary model optimization. Specifically, given the historical observations of the multi-agent dynamical system characterized as a temporal graph, we first extract invariant observation embeddings via a temporal graph neural network. Then, we introduce a set of prototypical prompts and combine the interpolation of the context signal into a graph ODE framework as proxy models. To capture the temporal distribution shifts, we predict trajectories under each prompt and reintroduce the confident set selected by a vision language model as augmentation data of the corresponding proxy model. To improve the generalization under parameter-induced distribution shifts, we jointly learn the primary and proxy models through a coordinated bi-level optimization strategy. Finally, experimental results on multiple real-world system dynamics datasets validate the superiority of our PEACE compared to state-of-the-art methods.

In summary, our paper makes the following contributions: ❶ *New Perspective:* We study an underexplored yet practical problem of out-of-distribution system dynamics modeling and propose a set of prototypical environment-aware proxy models to solve the problem. ❷ *Novel Methodology:* Our PEACE not only accumulates the temporal generalization effects of diverse augmentations through prototypical prompt-based proxy models, but also improves the generalization under system distribution shifts via the coordinated bi-level optimization to enforce gradient alignment across proxy models. ❸ *Extensive Experiment:* We conduct comprehensive experiments on multiple real-world system dynamics datasets, and the results demonstrate the superiority of our PEACE.

## 2  PRELIMINARIES & PROBLEM DEFINITION

**Notations.** Given a multi-agent dynamical system, we characterize the observation trajectories as a temporal graph $\mathcal{G}^{1:T_{obs}} = \{\mathcal{G}^1, \ldots, \mathcal{G}^{T_{obs}}\}$, where $\mathcal{G}^t = \{\mathcal{V}, \boldsymbol{A}^t, \boldsymbol{X}^t\}$ denotes the interaction

graph at timestep $t$. Here, $\mathcal{V}$ denotes the agent set within the graph. The adjacency matrix $\boldsymbol{A}^t$ encodes interaction between agents with $w_{ij}^t$ representing the pairwise weights between agent $i$ and $j$. The corresponding feature matrix can be denoted as $\boldsymbol{X}^t$, where each row $\boldsymbol{x}_i^t \in \mathbb{R}^d$ represents the $d$-dimensional state of the agent $i$.

**Neural ODEs for Dynamical Systems.** Neural Ordinary Differential Equations (Neural ODEs) (Chen et al., 2018) provide a continuous-time formulation of deep learning models, which have proven effective in modeling dynamical systems (Huang et al., 2020; 2021; Luo et al., 2023; 2024). For each agent $i \in \mathcal{V}$ within the system, the latent state can be represented as $\boldsymbol{z}_i^t$ and the evolution of the agent states can be formulated as:

$$\frac{d\boldsymbol{z}_i^t}{dt} = f(\boldsymbol{z}_i^t, \{\boldsymbol{z}_j^t | j \in \mathcal{V}/i\}, \boldsymbol{A}^t), \; \boldsymbol{z}_i^T = \boldsymbol{z}_i^0 + \int_{t=0}^{T} f(\boldsymbol{z}_i^t, \{\boldsymbol{z}_j^t | j \in \mathcal{V}/i\}, \boldsymbol{A}^t), \quad (1)$$

where $f(\cdot)$ models the interaction dynamics for the agents. And we can obtain $\boldsymbol{z}_i^t$ via numerical solvers such as Runge-Kutta (Schober et al., 2019) and Leapfrog (Zhuang et al., 2021).

**Problem Definition.** We consider a dynamical system parameterized through governing equations with a set of system variables $\boldsymbol{\xi}$. Distinct choices of $\boldsymbol{\xi}$ could influence the underlying dynamical principles as $P(\mathcal{G}^{1:T_{obs}}|\boldsymbol{\xi})$ and induce the distribution shift such that the trajectories across training and test dynamical systems are different, namely $P_{tr}(\mathcal{G}^{1:T_{obs}}) \neq P_{te}(\mathcal{G}^{1:T_{obs}})$. The objective is to learn a robust model $g_{\boldsymbol{\theta}}^*(\cdot) = h^*(f^*(\cdot))$ that achieve good performance across environment:

$$g^* = \arg\min_{g_{\boldsymbol{\theta}}} \sup_{\boldsymbol{\xi}} \mathcal{R}(g|\boldsymbol{\xi}), \quad (2)$$

where $h(\cdot)$ denotes a function to predict the future trajectories $\boldsymbol{X}^{T_{obs}+1:T}$ and $\mathcal{R}(\cdot)$ represents the empirical risk of the predicted trajectories.

**Environment Gradient Matching.** Without loss of generality, let $\mathcal{E} = \{e_1, e_2\}$ denote the collection of two source environments, the training loss of samples from the collection can be:

$$\mathcal{L} = \alpha_1 \mathcal{L}_1 + \alpha_2 \mathcal{L}_2, \; \mathcal{L}_i = -\mathbb{E}_e[\log h(y|\boldsymbol{z}; \boldsymbol{\theta})], \quad (3)$$

where $\alpha_s$ is the proportion of training samples drawn from environment $e \in \{e_1, e_2\}$, and $y$ denotes the label of the future trajectories. Thus, each training step depends on the loss gradient:

$$\nabla_{\boldsymbol{\theta}} \mathcal{L} = \alpha_1 \nabla_{\boldsymbol{\theta}} \mathcal{L}_1 + \alpha_2 \nabla_{\boldsymbol{\theta}} \mathcal{L}_2 = -\alpha_1 \mathbb{E}_{e=e_1} [\nabla_{\boldsymbol{\theta}} \log h(y|\boldsymbol{z}; \boldsymbol{\theta})] - \alpha_2 \mathbb{E}_{e=e_2} [\nabla_{\boldsymbol{\theta}} \log h(y|\boldsymbol{z}; \boldsymbol{\theta})], \quad (4)$$

where derivatives can be linearly decomposed across environments. We expect the prediction remain invariant across environments, namely $h(y|\boldsymbol{z}) = h(y|f(\boldsymbol{c}, \boldsymbol{p}_1|\boldsymbol{\xi})) = h(y|f(\boldsymbol{c}, \boldsymbol{p}_2|\boldsymbol{\xi}))$ under parameter $\boldsymbol{\xi}$, where $\boldsymbol{c}$ and $\boldsymbol{p}$ denote the invariant and specific embeddings. Thus, $\mathbb{E}_{e=e_1}[u(h(y|\boldsymbol{z}))] = \mathbb{E}_{e=e_2}[u(h(y|\boldsymbol{z}))]$ for any function $u$. This implies that, in expectation, both the loss $\mathcal{L}_e$ and its corresponding gradients $\nabla_{\boldsymbol{\theta}} \mathcal{L}_e$ are required to be consistent across environments.

## 3 THE PROPOSED PEACE

In this section, we present PEACE, a framework for generalized system dynamics modeling, which consists of a primary model to learn invariant observation embeddings and employs prototypical prompts as environment-aware proxy models. Figure 2 provides an overview of the framework and we present the details of each component below.

### 3.1 TEMPORAL GRAPH NEURAL NETWORKS FOR PRIMARY CONTEXT EXPLORATION

Given the trajectories of the dynamical system with $N$ agents, we construct a temporal graph to describe spatial and temporal correlations. The adjacency matrix $\boldsymbol{A}^{1:T_{obs}} \in \mathbb{R}^{NT_{obs} \times NT_{obs}}$ can be:

$$\boldsymbol{A}^{1:T_{obs}}(i^t, j^{t'}) = \begin{cases} w_{ij}^t, & t = t', \\ 1, & i = j, t' = t+1, \\ 0, & \text{otherwise}, \end{cases} \quad (5)$$

where $i^t$ is the observation of agent $i$ as timestep $t$ and $w_{ij}^t$ denotes the correlation weights. Based on the constructed temporal graph, we capture the spatio-temporal context to learn invariant observation

Figure 2: An overview of our PEACE. We leverage a temporal graph neural network as the primary model to explore the invariant context. Then, to model distribution shifts, we introduce a set of prototypical prompts as proxy models. Finally, the model is trained via a coordinated bi-level optimization strategy to enforce gradient alignment across proxy models.

embeddings. Specifically, with $\boldsymbol{h}_i^{t,l}$ denoted as the representation of $i^t$ at the $l$-th layer, we employ a primary model with the message-passing mechanism to learn invariant observation embeddings. The interaction score between node $i^t$ and $j^{t'}$ with temporal graph can be computed as:

$$\alpha^l(i^t, j^{t'}) = \frac{\boldsymbol{A}^{1:T_{obs}}(i^t, j^{t'})}{\sqrt{d}} (\boldsymbol{W}_{query} \hat{\boldsymbol{h}}_i^{t',l}) \star (\boldsymbol{W}_{key} \hat{\boldsymbol{h}}_j^{t',l}), \tag{6}$$

where $\boldsymbol{W}_{query}, \boldsymbol{W}_{key} \in \mathbb{R}^{d \times d}$ denote the query, key matrices and $\star$ computes the cosine similarity between two vectors. Then we aggregate from the neighbors $\mathcal{N}(i^t)$ for $i^t$ with the interaction score:

$$\boldsymbol{h}_i^{t,l+1} = \boldsymbol{h}_i^{t,l} + \sigma\Big( \sum_{j^{t'} \in \mathcal{N}(i^t)} \alpha^l(i^t, j^{t'}) \boldsymbol{W}_{value} \hat{\boldsymbol{h}}_j^{t',l} \Big), \tag{7}$$

where $\boldsymbol{W}_{value} \in \mathbb{R}^{d \times d}$ is the value matrix and $\sigma(\cdot)$ is an activation function. We provide the temporal information to update the observation embedding, i.e., $\hat{\boldsymbol{h}}_i^{t,l} = \boldsymbol{h}_i^{t,l} + \text{TE}(t)$ with $\text{TE}(t)[2i] = \sin(\frac{t}{10000^{2i/d}})$ and $\text{TE}(t)[2i + 1] = \cos(\frac{t}{10000^{2i/d}})$. By stacking $L$ layers, we denote the final invariant observation embedding context as $\boldsymbol{c}_i^t = \boldsymbol{h}_i^{t,L}$.

## 3.2 ENVIRONMENT-AWARE DYNAMIC PROMPTING FOR PROXY MODELS

Since dynamical systems often exhibit diverse distribution shifts, we fix the observation embedding $\boldsymbol{C}^t$ and incorporate distinct time-evolving prompts $\boldsymbol{P}^t$ as proxy models to obtain the final agent states $\boldsymbol{Z}^t$ for the system dynamics modeling under distribution shift, formulated as:

$$\boldsymbol{Z}^t = \phi([\boldsymbol{C}^t, \boldsymbol{P}^t]), \ \boldsymbol{C}^t \perp \boldsymbol{P}^t, \tag{8}$$

where $\phi(\cdot)$ denotes the downstream temporal decoder. We aim to utilize a set of $K$ prototypical prompts to indicate the distribution shift under diverse environments. Specifically, given the system parameters, we encode them with $\boldsymbol{u} = \phi_{PE}(\boldsymbol{\xi})$ and concatenate the embedding with the initial states of each agent. Afterward, we employ an attention mechanism to generate the prompt context, which is then used to initialize the prompt embeddings:

$$\boldsymbol{e}_i = \boldsymbol{u} \odot \boldsymbol{x}_i, \boldsymbol{E}^{l+1} = \phi_{SA}^l(\boldsymbol{E}^l), \tag{9}$$

where $\phi_{SA}^l(\cdot)$ denote the self-attention function to exploit prompt context, $\boldsymbol{E}^{k,0}$ is constructed by stacking $\{\boldsymbol{e}_i\}_{i=1}^N$ and $\odot$ is the Hadamard product. By stacking $L$ layers, we retrieve the initial prompt from the prompt context for each query agent states:

$$\boldsymbol{p}_i^{k,0} = \text{softmax}\left( \frac{\left(\boldsymbol{W}_{query'}^k \phi_{OE}(\boldsymbol{x}_i)\right) \star \left(\boldsymbol{W}_{key'}^k \boldsymbol{E}^L\right)}{\sqrt{d}} \right) \cdot \boldsymbol{W}_{value'}^k \boldsymbol{E}^L, \tag{10}$$

where $\phi_{OE}(\cdot)$ is the feed-forward network (FFNs), and $\boldsymbol{p}_i^{k,0}$ serves as the initial states of the prompt.

The updating rule of the $k$-th prototypes for agent $i$ is defined as:

$$f_s^k(\boldsymbol{p}_i^{k,t}, \{\boldsymbol{p}_j^{k,t}|j \in \mathcal{V}/i\}, \boldsymbol{A}^t) = \psi_a^k\big(\sum_{j \in \mathcal{N}^t(i)} \psi_r^k([\boldsymbol{p}_i^{k,t}, \boldsymbol{p}_j^{k,t}])\big), \quad (11)$$

where $\mathcal{N}^t(i)$ denotes the neighbors of agent $i$ defined by the graph structure $\boldsymbol{A}^t$, $\psi_a^k(\cdot)$ and $\psi_r^k(\cdot)$ are the relation learning and feature aggregation function to determine the $k$-th prompt evolution. We further interpolate the observation sequence $\boldsymbol{X}^{1:T_{obs}}$ to obtain the continuous states $\boldsymbol{X}^t$ at any timestep and incorporate them with the attention mechanism to adjust the evolution process:

$$\frac{d\boldsymbol{p}_i^{k,t}}{dt} = \psi_a^k\big(\sum_{j \in \mathcal{N}^t(i)} \text{softmax}\left(\frac{(\boldsymbol{W}_{query'}^k \phi_{OE}(\boldsymbol{x}_i^t)) \star (\boldsymbol{W}_{key'}^k \boldsymbol{p}_j^{k,t})}{\sqrt{d}}\right) \cdot \psi_r^k([\boldsymbol{p}_i^{k,t}, \boldsymbol{p}_j^{k,t}])\big). \quad (12)$$

We also decouple the learned observation and prompt embedding to enhance the invariance of the model under temporal distribution. To achieve this, we minimize the mutual information between two embeddings, which can be formulated as:

$$\mathcal{L}_{MI}^k = -\frac{1}{|\mathcal{P}|}\sum_{(\boldsymbol{c}_i^t, \boldsymbol{p}_i^{k,t}) \in \mathcal{P}} sp(-D(\boldsymbol{c}_i^t, \boldsymbol{p}_i^{k,t})) + \frac{1}{|\mathcal{N}||\mathcal{S}|}\sum_{(\boldsymbol{c}_i^t, \boldsymbol{p}_j^{k,t}) \notin \mathcal{P}} sp(-D(\boldsymbol{c}_i^t, \boldsymbol{p}_j^{k,t})), \quad (13)$$

where the loss is estimated from discriminator $D(\cdot)$ using all pairs $(\boldsymbol{c}_i^t, \boldsymbol{p}_i^{k,t})$ as positive set $\mathcal{P}$ and other pairs as $\mathcal{S}$, and $sp(x) = \log(1 + e^x)$.

## 3.3 GUIDED TRAJECTORY SIMULATION FOR DATA AUGMENTATION

Dynamic systems often exhibit non-stationary behaviors due to the underlying dynamics evolving and inducing discrepancies over time. To improve robustness against such shifts, we generate unseen trajectories from each proxy model as augmented data to simulate temporal variations. Specifically, we concatenate the observation embedding and the time-evolving prompt as follows:

$$\boldsymbol{z}_i^{k,t} = [\boldsymbol{c}_i^t, \boldsymbol{p}_i^{k,t}]. \quad (14)$$

And the updated embeddings are fed into the decoder $\phi(\cdot)$ to generate the subsequent trajectory $\hat{\boldsymbol{X}}^{T_{obs}+1:T}$. Since the predictions may still be affected by temporal distribution shifts and spurious patterns, we further incorporate a vision language model (VLM) to jointly reason over the visual dynamics and semantic context, and evaluate whether the generated trajectories align with plausible system behaviors. Specifically, the predicted trajectories are transformed into visual plots $\tilde{\boldsymbol{V}}$ and paired with textual descriptions $\tilde{\boldsymbol{X}}^{T_{obs}+1:T}$, which are then provided as inputs to a set of $n$ VLM agents $\mathcal{M} = \{\mathcal{M}_1, \mathcal{M}_2, \ldots, \mathcal{M}_n\}$. And we obtain multiple responses as:

$$\{\mathcal{R}_m\} = \{\mathcal{M}_m(\tilde{\boldsymbol{V}}, \tilde{\boldsymbol{X}}^{T_{obs}+1:T})\}_{m=1}^n. \quad (15)$$

We employ the *self-justification* which takes the vision and textual description input as well as the corresponding response to generate the final assessment:

$$\mathcal{J} = \text{Self-Justify}(\tilde{\boldsymbol{V}}, \tilde{\boldsymbol{X}}^{T_{obs}+1:T}, \{\mathcal{R}_m\}_{m=1}^n), \quad (16)$$

where the predicted unseen trajectory serves as the augmented data for each proxy model according to $\mathcal{J}$, and is subsequently utilized to update the model parameters during training.

## 3.4 BI-LEVEL OPTIMIZATION WITH GRADIENT COORDINATION

To further improve the generalization of the model under parameter-induced distribution shifts, we propose the following alternative optimization schema to effectively explore both the proxy model parameters $\{\boldsymbol{\theta}_{\boldsymbol{p}^k}\}_{k=1}^K$ and primary model parameters $\boldsymbol{\theta}_{\boldsymbol{c}}$.

**Lower-level Optimization.** For each proxy model, we fix the primary model parameters $\boldsymbol{\theta}_{\boldsymbol{c}}$ and update specific parameters $\boldsymbol{\theta}_{\boldsymbol{p}^k}$ within each proxy model. Given the predicted observations under both proxy models and their ground-truth, we minimize the mean squared error (MSE) loss:

$$\mathcal{L}_{MSE}^k = \sum_{t=T_{obs}+1}^T \|\hat{\boldsymbol{X}}^t - \boldsymbol{X}^t\|. \quad (17)$$

The prototypical prompt $\boldsymbol{p}^k$ and its parameters $\boldsymbol{\theta}_{\boldsymbol{p}^k}$ within each proxy model can be updated as:

$$\boldsymbol{\theta}_{\boldsymbol{p}^k}^{a+1} = \boldsymbol{\theta}_{\boldsymbol{p}^k}^a - \gamma_{\boldsymbol{p}} \nabla_{\boldsymbol{\theta}_{\boldsymbol{p}^k}} (\mathcal{L}_{MSE}^k(\boldsymbol{c}, \boldsymbol{p}^k) + \mathcal{L}_{MI}^k(\boldsymbol{c}, \boldsymbol{p}^k)), \tag{18}$$

where $\gamma_{\boldsymbol{p}}$ is the learning rate, with the prototypical prompt embedding updates $n_a$ iterations.

**Upper-level Optimization.** Since the agreement of the gradient $\nabla \mathcal{L}_e$ across environments indicates invariant learning, the updated prototypical prompts $\tilde{\boldsymbol{p}}^k$ are leveraged to assess gradient coordination and guide the update of the primary model. We assume the less the model depends on the environments, the more similar the expected gradient among prototypical prompts. Inspired by the strategy of simulated annealing (Kirkpatrick et al., 1983; Ballas & Diou, 2025), we iteratively add random noise to the prompt embeddings and explore the point with model gradient agreement:

$$(\tilde{\boldsymbol{p}}^k)' = \tilde{\boldsymbol{p}}^k + \mathcal{U}(-\rho, \rho), \tag{19}$$

where $\mathcal{U}$ denotes the multivariate uniform distribution within the range $[-\rho, \rho]$. And we measure the gradient similarity among proxy models as:

$$\text{Grad-Sim} = \min_{\substack{1 \leq k_1 \leq K, 1 \leq k_2 \leq K \\ k_1 \neq k_2}} \left( \frac{\boldsymbol{g}_{k_1}^T \cdot \boldsymbol{g}_{k_2}}{\|\boldsymbol{g}_{k_1}\| \|\boldsymbol{g}_{k_2}\|} \right), \tag{20}$$

where $\boldsymbol{g}_k$ serves as the gradient of the observation embeddings within the $k$-th proxy. By selecting the point with the highest gradient agreement and the lowest loss, the Pareto front can be refined through several iterative steps. The parameters of the primary model $\boldsymbol{\theta}_{\boldsymbol{c}}$ can be updated by:

$$\boldsymbol{\theta}_c^{b+1} = \boldsymbol{\theta}_c^b - \gamma_{\boldsymbol{c}} \nabla (\mathcal{L}_{MSE}(\boldsymbol{c}, \tilde{\boldsymbol{p}}')), \ \tilde{\boldsymbol{p}}' = \sum_{k=1}^K (\tilde{\boldsymbol{p}}^k)'. \tag{21}$$

Note that here we integrate all these proxy models via parameter averaging, enabling the ensemble model to accumulate the effects of diverse augmentations (Rame et al., 2022; Cho et al., 2025).

## 3.5 THEORETICAL ANALYSIS

In this section, we present a theoretical analysis of the proposed PEACE framework. For simplicity of notation, we impose two simplifying assumptions: ❶ The functions $\psi_a^k(\cdot)$ and $\psi_r^k(\cdot)$ are linear; and ❷ The softmax term can be represented by a weight matrix $w^k$,

$$w^k = \text{softmax} \left( \frac{\left( \boldsymbol{W}_{Q'}^k \phi_{OE}(\boldsymbol{x}_i^t) \right) \star \left( \boldsymbol{W}_{K'}^k \boldsymbol{p}_j^{k,t} \right)}{\sqrt{d}} \right). \tag{22}$$

Let $\boldsymbol{p}^k = [\boldsymbol{p}_1^k, \boldsymbol{p}_2^k, \ldots, \boldsymbol{p}_N^k]$ denote the prompt matrix for all agents, and let $\boldsymbol{A}_t^k$ be the adjacency at time $t$. Under these assumptions, the ODE governing the $k$-th prototypical prompt simplifies to:

$$\frac{d\boldsymbol{p}^{k,t}}{dt} = \boldsymbol{A}_t^k w^k \boldsymbol{p}^{k,t}. \tag{23}$$

Since the overall prompt is the sum of all $K$ prototypical prompts, we obtain:

$$\frac{d\boldsymbol{p}^t}{dt} = \sum_{k=1}^K \frac{d\boldsymbol{p}^{k,t}}{dt} = \sum_{k=1}^K \boldsymbol{A}_t^k w^k \boldsymbol{p}^{k,t}. \tag{24}$$

To facilitate the proof of the main theorem, we assume that the adjacency matrices are time-invariant, i.e., $\boldsymbol{A}_t^k = \boldsymbol{A}^k$. Let $\boldsymbol{A}$ denote the block-diagonal matrix whose diagonal blocks are given by $w^k \boldsymbol{A}^k$. Under this notation, the ODE for the overall prompt dynamics becomes $\frac{d\boldsymbol{p}^t}{dt} = \boldsymbol{A}\boldsymbol{p}^t$. This time-invariance assumption is reasonable in many practical scenarios where the interaction among agents remains relatively stable. Moreover, we assume that the *true* system dynamics are governed by $\frac{d\boldsymbol{p}^t}{dt} = \boldsymbol{A}^*\boldsymbol{p}^t$, where $\boldsymbol{A}^*$ is the ground-truth dynamics matrix. With coordinated bi-level optimization, the learned dynamics matrix $\boldsymbol{A}$ provides a closer approximation to $\boldsymbol{A}^*$, whereas without coordinated optimization, the learned matrix is denoted by $\tilde{\boldsymbol{A}}$, which may deviate from $\boldsymbol{A}^*$. This

Table 1: Mean Squared Error (MSE) $\times 10^{-2}$ on *Springs* dataset, where $q$ denotes position and $v$ denotes velocity error.

| Prediction length | 12 (ID) | | 24 (ID) | | 36 (ID) | | 12 (OOD) | | 24 (OOD) | | 36 (OOD) | |
| Methods | $q$ | $v$ | $q$ | $v$ | $q$ | $v$ | $q$ | $v$ | $q$ | $v$ | $q$ | $v$ |
|---|---|---|---|---|---|---|---|---|---|---|---|---|
| LSTM | 0.287 | 0.920 | 0.659 | 2.659 | 1.279 | 5.729 | 0.474 | 1.157 | 0.938 | 2.656 | 1.591 | 5.223 |
| GRU | 0.394 | 0.597 | 0.748 | 1.856 | 1.248 | 3.446 | 0.591 | 0.708 | 1.093 | 1.945 | 1.671 | 3.423 |
| NODE | 0.157 | 0.564 | 0.672 | 2.414 | 1.608 | 6.232 | 0.228 | 0.791 | 0.782 | 2.530 | 1.832 | 6.009 |
| LG-ODE | 0.077 | 0.268 | 0.155 | 0.513 | 0.527 | 2.143 | 0.088 | 0.299 | 0.179 | 0.562 | 0.614 | 2.206 |
| MP-NODE | 0.076 | 0.243 | 0.171 | 0.456 | 0.600 | 1.737 | 0.094 | 0.249 | 0.212 | 0.474 | 0.676 | 1.716 |
| SocialODE | 0.069 | 0.260 | 0.129 | 0.510 | 0.415 | 2.187 | 0.079 | 0.285 | 0.153 | 0.570 | 0.491 | 2.310 |
| HOPE | 0.070 | 0.176 | 0.456 | 0.957 | 2.475 | 5.409 | 0.076 | 0.221 | 0.515 | 1.317 | 2.310 | 5.996 |
| Pioneer | 0.059 | 0.150 | 0.389 | 0.817 | 2.114 | 4.621 | 0.064 | 0.189 | 0.439 | 1.125 | 1.973 | 5.122 |
| PGODE | 0.035 | 0.124 | 0.070 | 0.262 | 0.296 | 1.326 | 0.047 | 0.138 | 0.088 | 0.291 | 0.309 | 1.337 |
| **PEACE** (ours) | **0.018** | **0.119** | **0.031** | **0.199** | **0.224** | **0.704** | **0.040** | **0.136** | **0.086** | **0.208** | **0.279** | **0.753** |

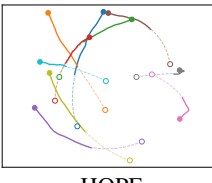 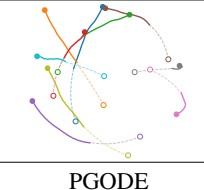 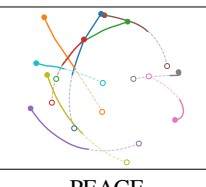 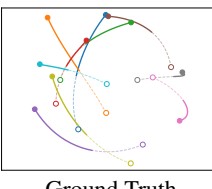

| HOPE | PGODE | PEACE | Ground Truth |

Figure 3: Visualization of different methods on *Springs*. Dashed lines denote observed trajectories, solid lines represent predicted trajectories with a length of 12 steps.

deviation can be characterized by the spectral differences between the matrices, leading to significant prediction errors for models without our coordinated optimization strategy (Van Loan, 1977). To show how coordinated optimization mitigates this, we now provide a bound on the prediction error for our proposed model:

**Theorem 3.1.** *Consider the linear ODE systems*

$$\frac{d\boldsymbol{p}^t}{dt} = \boldsymbol{A}\,\boldsymbol{p}^t, \qquad \frac{d\boldsymbol{p}^{*t}}{dt} = \boldsymbol{A}^*\,\boldsymbol{p}^{*t}, \tag{25}$$

*subject to the common initial condition $\boldsymbol{p}^0 = \boldsymbol{p}^{*0} = \boldsymbol{p}_0$. Then, for every $t \geq 0$,*

$$\|\boldsymbol{p}^t - \boldsymbol{p}^{*t}\|_2 \;\leq\; t\exp\Big(t\big(\|\boldsymbol{A}\|_2 + \|\boldsymbol{A}^*\|_2\big)\Big)\,\|\boldsymbol{A} - \boldsymbol{A}^*\|_2\,\|\boldsymbol{p}_0\|_2. \tag{26}$$

Theorem 3.1 provides an upper bound on the difference between the learned dynamics and the true dynamics in terms of the spectral norm of the difference between their respective dynamics matrices. The bound indicates that as the learned dynamics matrix $\boldsymbol{A}$ approaches the true dynamics matrix $\boldsymbol{A}^*$ (i.e., $\|\boldsymbol{A} - \boldsymbol{A}^*\|_2$ decreases), the discrepancy between the learned and true dynamics diminishes. This result underscores the importance of accurately estimating the dynamics matrix to ensure that the model can effectively capture the underlying system behavior, thereby enhancing its generalization ability across different environments. The proof of Theorem 3.1 is provided in Appendix C.1.

## 4 EXPERIMENT

### 4.1 EXPERIMENTAL SETTINGS

***Datasets & Baselines.*** We evaluate PEACE on four benchmark datasets covering both physical (Springs, Charged (Schlichtkrull et al., 2018)) and molecular (5AWL, 2N5C (Luo et al., 2024)) dynamics. Each trajectory is divided into a conditioning segment, which encodes historical observations for initializing object- and system-level states, and a prediction segment, which serves as the ground-truth target for forecasting. By varying the prediction length, we construct scenarios that test the model's ability to capture both short- and long-term dynamics. Dataset details are provided in Appendix E.1. We compare PEACE with a comprehensive set of representative approaches, including sequence models (LSTM (Hochreiter & Schmidhuber, 1997), GRU (Cho et al., 2014)), neural ODEs (NODE (Chen et al., 2018), LG-ODE (Huang et al., 2020), MP-NODE (Chen et al., 2022b), SocialODE (Wen et al., 2022)), and recent dynamics models (HOPE (Luo et al., 2023), PGODE (Luo et al., 2024), Pioneer (Sun et al., 2025)). Full descriptions are provided in Appendix E.2. The implementation details, including training schedules and data preprocessing, are reported in Appendix E.3.

Table 2: Mean Squared Error (MSE) $\times 10^{-2}$ on *5AWL* dataset.

| Prediction length | 12 (ID) | | | 24 (ID) | | | 12 (OOD) | | | 24 (OOD) | | |
|---|---|---|---|---|---|---|---|---|---|---|---|---|
| Methods | $q_x$ | $q_y$ | $q_z$ | $q_x$ | $q_y$ | $q_z$ | $q_x$ | $q_y$ | $q_z$ | $q_x$ | $q_y$ | $q_z$ |
| LSTM | 0.417 | 0.339 | 0.395 | 0.435 | 0.444 | 0.398 | 0.478 | 0.417 | 0.446 | 0.515 | 0.521 | 0.454 |
| GRU | 0.436 | 0.286 | 0.283 | 0.529 | 0.384 | 0.399 | 0.513 | 0.366 | 0.378 | 0.600 | 0.472 | 0.535 |
| NODE | 0.399 | 0.329 | 0.248 | 0.467 | 0.433 | 0.325 | 0.439 | 0.413 | 0.280 | 0.573 | 0.538 | 0.403 |
| LG-ODE | 0.282 | 0.280 | 0.256 | 0.372 | 0.394 | 0.341 | 0.335 | 0.354 | 0.350 | 0.461 | 0.476 | 0.454 |
| MP-NODE | 0.263 | 0.302 | 0.273 | 0.358 | 0.415 | 0.348 | 0.306 | 0.389 | 0.335 | 0.427 | 0.508 | 0.442 |
| SocialODE | 0.248 | 0.272 | 0.247 | 0.332 | 0.395 | 0.339 | 0.298 | 0.351 | 0.316 | 0.424 | 0.479 | 0.415 |
| HOPE | 0.232 | 0.257 | 0.244 | 0.349 | 0.381 | 0.341 | 0.258 | 0.352 | 0.295 | 0.454 | 0.504 | 0.400 |
| Pioneer | 0.228 | 0.249 | 0.208 | 0.298 | 0.325 | 0.291 | 0.220 | 0.301 | 0.252 | 0.388 | 0.431 | 0.342 |
| PGODE | 0.209 | 0.234 | 0.209 | 0.291 | 0.338 | 0.290 | 0.221 | 0.310 | 0.259 | 0.337 | 0.433 | 0.361 |
| **PEACE** (ours) | **0.203** | **0.230** | **0.144** | **0.254** | **0.299** | **0.266** | **0.213** | **0.247** | **0.161** | **0.266** | **0.370** | **0.273** |

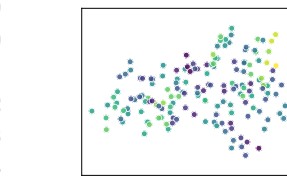 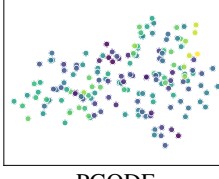 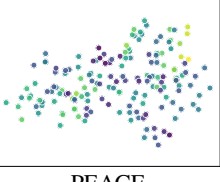 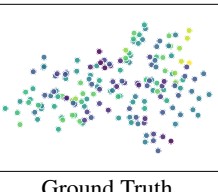

| HOPE | PGODE | PEACE | Ground Truth |

Figure 4: Visualization of different methods on *5AWL*. The figure illustrates the molecular movement trajectories at the 24th predicted step.

## 4.2 PERFORMANCE ON PHYSICAL DYNAMICS SIMULATIONS

***Performance Comparison.*** In the Springs datasets, each sample contains 10 particles moving within a two-dimensional box, where collisions may occur in the absence of external forces. The task is to forecast future positions $q$ and velocities $v$. Results across different prediction horizons are summarized in Table 1. Two key observations can be drawn: ❶ ODE-based approaches consistently surpass discrete models, highlighting the effectiveness of continuous-time formulations in capturing dynamics and mitigating error accumulation; and ❷ PEACE achieves the best overall performance, yielding an average MSE reduction of 22.05% compared with PGODE and 19% in OOD settings. The superiority of PEACE can be attributed to three factors: (i) a dual proxy and primary model that decouples contextual encoding from dynamics modeling, (ii) guided trajectory simulation for data augmentation, and (iii) coordinated optimization that promotes cross-domain generalization. Additional comparative results are reported in Appendix E.4.

***Visualization.*** Figure 3 presents qualitative comparisons among PEACE, HOPE, PGODE, and the ground truth on the Springs. From the predicted trajectories (solid lines), PEACE closely tracks the ground truth across all particles, whereas HOPE and PGODE exhibit pronounced deviations, particularly in regions involving sharp directional changes. Notably, PEACE is able to accurately capture collective motion patterns and interactions among multiple particles, maintaining trajectory coherence even in complex configurations. Moreover, its predictions remain stable across different forecast horizons, further validating the robustness and accuracy of PEACE in modeling complex particle dynamics. More visualization results are supplemented in the appendix E.6.

## 4.3 PERFORMANCE ON MOLECULAR DYNAMICS SIMULATIONS

***Performance Comparison.*** We further evaluate PEACE on the molecular dynamics (MD) dataset. Each sample corresponds to a trajectory where atom motions follow Langevin dynamics with solvent parameters varying across different simulations. The task is to forecast atomic positions along the three spatial coordinates $q_x$, $q_y$, and $q_z$. Table 2 summarizes the results under different prediction horizons. Despite the higher complexity of molecular dynamics compared with physical systems, PEACE consistently achieves the lowest MSE across both ID and OOD settings. These results highlight the strong capability of PEACE in modeling intricate interaction rules and generalizing across diverse solvent conditions. Results on additional datasets are provided in the Appendix E.4.

***Visualization.*** Figure 4 presents qualitative comparisons among PEACE, HOPE, PGODE, and the ground truth. Owing to the large number of atoms, we display snapshots of predicted positions at the 24th prediction step. PEACE closely reproduces the underlying molecular dynamics, maintaining stable trajectories without noticeable drift, whereas HOPE and PGODE exhibit larger deviations. Notably, PEACE preserves intricate inter-atomic interactions, capturing local structural correlations

Table 3: Ablation study on *Springs*, *Charged* and *5AWL* with a prediction length of 24.

| Dataset | *Springs* (ID) | | *Springs* (OOD) | | *Charged* (ID) | | *Charged* (OOD) | | *5AWL* (ID) | | | *5AWL* (OOD) | | |
|---|---|---|---|---|---|---|---|---|---|---|---|---|---|---|
| Variable | $q$ | $v$ | $q$ | $v$ | $q$ | $v$ | $q$ | $v$ | $q_x$ | $q_y$ | $q_z$ | $q_x$ | $q_y$ | $q_z$ |
| W/O PROXY MODELS | 0.216 | 0.336 | 0.201 | 0.317 | 2.407 | 2.522 | 2.768 | 2.827 | 0.604 | 0.719 | 0.561 | 0.680 | 0.747 | 0.563 |
| W/O BI-LEVEL OPTIMIZATION | 0.054 | 0.254 | 0.108 | 0.451 | 2.756 | 2.408 | 2.955 | 2.456 | 0.316 | 0.383 | 0.800 | 0.339 | 0.388 | 0.826 |
| W/O SIMULATION | 0.048 | 0.343 | 0.193 | 0.317 | 2.008 | 1.947 | 2.427 | 2.337 | 0.269 | 0.385 | 0.528 | 0.285 | 0.394 | 0.539 |
| W/O VLM | 0.043 | 0.286 | 0.130 | 0.289 | 2.234 | 2.174 | 2.430 | 2.328 | 0.248 | 0.300 | 0.472 | 0.273 | 0.375 | 0.335 |
| **PEACE (ours)** | **0.031** | **0.107** | **0.086** | **0.172** | **1.893** | **1.814** | **2.337** | **2.252** | **0.204** | **0.269** | **0.207** | **0.266** | **0.371** | **0.273** |

effectively. These visualizations reinforce the quantitative results, further demonstrating the robustness and generalization ability of PEACE in MD modeling.

## 4.4 ABLATION STUDIES

To disentangle the contribution of individual components in PEACE, we perform ablation experiments by systematically removing or modifying key modules. We consider four variants: PEACE w/o Proxy Models, PEACE w/o Bi-level Optimization, PEACE w/o Simulation, and PEACE w/o VLM. As shown in Table 3, removing Proxy Models, which serve as environment-aware prototypical prompts, leads to a substantial performance drop, especially in velocity prediction and OOD generalization, underscoring their role in capturing distribution shifts and providing robust context for the primary model. Omitting Bi-level Optimization, which coordinates gradient updates between proxy and primary models, also results in significant degradation, highlighting its importance for hierarchical adaptation and cross-environment generalization. The Simulation module, responsible for generating augmented trajectories under each proxy model to mitigate temporal evolution-induced feature distortion, mainly stabilizes training on *Charged*. Finally, removing VLM-based self-justification, which evaluates the plausibility of generated trajectories using visual and semantic cues, proves crucial for the high-dimensional dataset, confirming its utility in validating proxy model outputs. Overall, these results demonstrate that each component contributes meaningfully to the robustness and accuracy of PEACE.

## 4.5 PARAMETER SENSITIVITY.

***Condition length.*** We vary the condition length from 3 to 15 under prediction horizons of 12 and 24 on the Springs and 5AWL datasets. As shown in Figure 5, performance generally decreases as condition length grows, but saturates

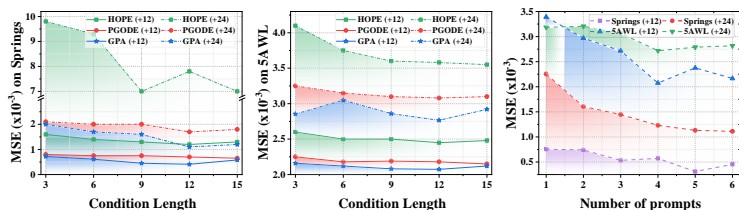

Figure 5: Parameter sensitivity analysis on Springs and 5AWL.

once the length exceeds 12, indicating diminishing returns from additional historical information. Across all settings, PEACE consistently outperforms HOPE and PGODE, demonstrating its robustness. ***Number of prompts.*** We also investigate the effect of prompt number. As shown in Figure 5, increasing prompts improves performance initially, but gains diminish and eventually saturate as training complexity rises. This highlights a trade-off between enhanced capacity and optimization difficulty, with PEACE remaining superior to baselines under all configurations.

## 5 CONCLUSION

In this work, we tackle the challenge of multi-agent dynamical systems modeling and introduce a novel framework, PEACE, which integrates prototypical proxy models to simulate environment information while keeping the primary model fixed. Specifically, PEACE first utilizes a temporal neural network to learn invariant observation embeddings. Then, a range of prototypical prompts is introduced to model distinct distribution shifts with graph ODEs and incorporated with observation embedding as the proxy models. Furthermore, we select high-confidence unseen trajectories generated from proxy models to mitigate temporal distribution shifts and introduce coordinated bi-level optimization to ensure stable prompt adaptation. Comprehensive experiments across diverse physical and molecular dynamics benchmarks demonstrate that PEACE achieves superior accuracy and generalization compared to existing approaches.

## REPRODUCIBILITY STATEMENT

We ensure the reproducibility of our work as follows. The main components of the proposed PEACE framework are detailed in Sections 3.1–3.5, including all mathematical derivations. Experimental settings, including hyperparameters, hardware specifications, datasets, and the complete training pipeline, are provided in Section 4.1 and Appendices E. An anonymous link to our implementation is provided at the end of abstract to facilitate independent verification.

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

## A  LARGE LANGUAGE MODEL (LLM) USAGE STATEMENT

We use the LLM as a general-purpose assistant tool. Specifically, the LLM assists in (i) checking grammar and improving clarity of text descriptions, and (ii) suggesting alternative phrasings for some sections. No parts of the paper are generated entirely by the LLM. All research ideas, experiments, model designs, and results are conceived, implemented, and analyzed solely by the authors. The LLM does not contribute to the development of the methodology, experiments, or analysis presented in this paper. We confirm that the use of the LLM is limited to minor writing support and does not constitute a substantive contribution that would qualify it as a co-author.

## B  RELATED WORK

We review existing works on three lines of fields: 1) Multi-agent Dynamical System Modeling, 2) Out-of-distribution Generalization, 3) Prompt Learning.

**Multi-agent Dynamical System Modeling.** Multi-agent dynamical systems provide a general framework for studying collective behavior across domains such as molecular dynamics and computational physics (Abeyruwan et al., 2023; Sun et al., 2023a; Yu et al., 2024). A central challenge lies in jointly modeling agent states and their evolving interaction structures (Kipf et al., 2018; Cini et al., 2025). Graph Neural Networks (GNNs) have been widely employed as discrete-time simulators (Huang et al., 2024; Bernárdez et al., 2023), effectively capturing temporal and relational dependencies via message passing. Extensions incorporate semantic reasoning (Yuan et al., 2021a), contrastive learning objectives (Yang et al., 2020), and equivariant architectures to enforce physical symmetries (Wu et al., 2023). However, GNN-based simulators generally assume regular sampling, suffer from error accumulation in long rollouts, and are sensitive to distribution shifts (Yu et al., 2021). To overcome these limitations, neural ordinary differential equations (ODEs) (Chen et al., 2018) have been introduced to parameterize continuous-time dynamics. Neural ODEs naturally handle irregular sampling and adaptive integration, and have demonstrated strong performance in time-series forecasting (Jin et al., 2023; Schirmer et al., 2022). Integrating ODEs with GNNs yields Graph ODEs (Zhang et al., 2022a), which unify spatial message passing with continuous temporal evolution, alleviating oversmoothing and improving interpretability. Further developments, such as HOPE (Luo et al., 2023), extend this paradigm to higher-order dynamics. Overall, the shift from learning discrete transition functions to approximating vector fields provides a more principled foundation for multi-agent dynamical system modeling.

**Out-of-distribution Generalization.** A central challenge in modeling multi-agent dynamical systems is out-of-distribution (OOD) generalization. Models trained under the i.i.d. assumption often fail when evaluated on shifted distributions (Fan et al., 2024; Xu et al., 2025). Such shifts can arise from changes in system parameters, temporal evolution, or environmental conditions, leading models to rely on spurious correlations rather than the true causal mechanisms of the dynamics. Recent research has explored three complementary directions. ❶ *Invariant learning* aims to extract features that remain stable across environments, as in CIGA (Chen et al., 2022c), StableGNN (Fan et al., 2024), and GPro (Xu et al., 2025). ❷ *Data-centric approaches*, including structural augmentations and contrastive learning (Feng et al., 2020), expand the training distribution to improve robustness. ❸ *Architectural and learning innovations*, such as adversarial training (Xue et al., 2021), inductive-bias designs (Li et al., 2025a), and test-time adaptation (Chen et al., 2022a), further enhance resilience to distribution shifts. Collectively, these efforts mark a transition from descriptive modeling toward causal-oriented reasoning, emphasizing invariant and causal features over spurious statistical patterns. In multi-agent dynamical systems, this is especially critical, as evolving states and dynamic environments can otherwise compromise long-term prediction and control.

**Prompt Learning.** Prompt learning adapts large pretrained models to downstream tasks by introducing task-specific prompts rather than fine-tuning the entire model. Initially proposed for natural language processing, this paradigm has been extended to other modalities such as graphs and time series. On graphs, prior studies explore both feature-space and token-based prompts: Graph Prompt Feature (GPF) introduces learned feature prompts for pretrained GNNs (Fang et al., 2023), while unified prompt tokens have been designed for multi-task GNNs (Sun et al., 2023b). In time-series forecasting, prompt-based adaptation has been leveraged to capture distribution shifts, either through GPT-style architectures (TEMPO) (Cao et al., 2024) or by prepending learned prototypes as

prefixes (Jin et al., 2024). Despite these advances, most existing methods rely on static prompts tied to fixed training environments. To overcome this limitation, recent research has explored dynamic prompt learning, such as online adaptation under distribution shifts (Xiao et al., 2025) and time-evolving prompt embeddings parameterized by Graph-ODEs (Wu et al., 2024). These directions highlight the emerging importance of prompt dynamics for enhancing robustness in non-stationary systems. Nevertheless, prompt learning in graph-based multi-agent dynamical systems remains largely underexplored, particularly under out-of-distribution conditions, leaving open opportunities for methods that explicitly couple prompt dynamics with system evolution.

## C THEORETICAL ANALYSIS

### C.1 PROOF OF THEOREM 3.1

*Proof of Theorem 3.1.* To begin, we first introduce a lemma from (Boyce & DiPrima, 2009).

**Lemma C.1.** *Consider that $\boldsymbol{A} \in \mathbb{R}^{n \times n}$ is a constant matrix and we have a first-order linear system*

$$\frac{d\mathbf{p}(t)}{dt} = \boldsymbol{A}\,\mathbf{p}(t), \qquad \mathbf{p}(0) = \mathbf{p}_0 \in \mathbb{R}^n, \tag{27}$$

*and then its unique solution is $\mathbf{p}(t) = e^{\boldsymbol{A}t}\mathbf{p}_0$.*

Then, we introduce another lemma from in (Higham, 2008).

**Lemma C.2.** *Considering $A, B \in \mathbb{R}^{n \times n}$ and denoting $\|\cdot\|$ as any sub-multiplicative matrix norm, i.e. $\|XY\| \leq \|X\|\|Y\|$ for all conformable matrices $X, Y$, we have*

$$\left\|e^A - e^B\right\| \leq \|A - B\| \frac{e^{\|A\|} - e^{\|B\|}}{\|A\| - \|B\|} \leq e^{\max\{\|A\|, \|B\|\}}\|A - B\|. \tag{28}$$

According to Lemma C.1, the solutions to our ODEs are formulated as :

$$\boldsymbol{p}^t = e^{\boldsymbol{A}t}\boldsymbol{p}_0, \qquad \boldsymbol{p}^{*t} = e^{\boldsymbol{A}^*t}\boldsymbol{p}_0. \tag{29}$$

Then, we calculate the difference between these two solutions at the time $t$ as follows:

$$\boldsymbol{p}^t - \boldsymbol{p}^{*t} = \left(e^{\boldsymbol{A}t} - e^{\boldsymbol{A}^*t}\right)\boldsymbol{p}_0. \tag{30}$$

By applying Lemma C.2 to both matrices in the equation, we obtain:

$$\left\|\boldsymbol{p}^t - \boldsymbol{p}^{*t}\right\|_2 \leq \left\|e^{\boldsymbol{A}t} - e^{\boldsymbol{A}^*t}\right\|_2 \|\boldsymbol{p}_0\|_2 \leq t\,e^{t\left(\|\boldsymbol{A}\|_2 + \|\boldsymbol{A}^*\|_2\right)}\|\boldsymbol{A} - \boldsymbol{A}^*\|_2\|\boldsymbol{p}_0\|_2. \tag{31}$$

This finishes our proof of Theorem 3.1. □

## D ALGORITHM

We summarize the learning algorithm of our PEACE in Algorithm 1.

---

**Algorithm 1** Prototypical Environment-aware Proxy with Coordinated Optimization (**PEACE** )

---

**Require:** Trajectories $\boldsymbol{X}^{1:T_{obs}}$, Adjacency matrices $\boldsymbol{A}^{1:T_{obs}}$, System params $\boldsymbol{\xi}$, Num prompts $K$, Learning rates $\gamma_{\boldsymbol{c}}, \gamma_{\boldsymbol{p}}$, Perturbation range $\rho$, Lower-level steps $n_a$, Annealing start step $A_s$, Annealing end step $A_e$, Annealing search steps $n_s$.

**Ensure:** Optimized model parameters $\boldsymbol{\theta_c}$ and $\{\boldsymbol{\theta_{p^k}}\}_{k=1}^{K}$.

1: Initialize parameters $\boldsymbol{\theta_c}$ (for primary model) and $\{\boldsymbol{\theta_{p^k}}\}_{k=1}^{K}$ (for proxy models).
2: **for** $t = 1$ to Total Training Steps **do**
3:     **Step 1: Primary Context Exploration**
4:     Compute invariant observation embeddings $\boldsymbol{C}^{1:T_{obs}} \leftarrow \phi_c(\boldsymbol{X}^{1:T_{obs}}, \boldsymbol{A}^{1:T_{obs}})$.
5:     Generate prompt context $\boldsymbol{E}^L$ from system parameters $\boldsymbol{\xi}$ via self-attention (Eq. 9).
6:     Initialize $K$ prototypical prompts $\{\boldsymbol{p}_i^{k,0}\}_{k=1}^{K}$ by attending to context $\boldsymbol{E}^L$ (Eq. 10).
7:     **Step 2: Lower-level Optimization (Proxy Model Training)**
8:     **for** $a = 1$ to $n_a$ **do**
9:         **for** $k = 1$ to $K$ **do**            ▷ Update each proxy model
10:            Evolve prompt $\boldsymbol{p}^{k,t}$ over time using Graph ODE with $\boldsymbol{\theta_{p^k}}$ (Eq. 12).
11:            Form combined state $\boldsymbol{z}^{k,t} \leftarrow [\boldsymbol{c}^t, \boldsymbol{p}^{k,t}]$.
12:            Predict trajectory $\hat{\boldsymbol{X}}^t$ with decoder using $\boldsymbol{z}^{k,t}$.
13:            Compute losses $\mathcal{L}_{MI}^k(\boldsymbol{c}^t, \boldsymbol{p}^{k,t})$ and $\mathcal{L}_{MSE}^k(\hat{\boldsymbol{X}}^t, \boldsymbol{X}^t)$ (Eq. 13, 17).
14:            Update prompt parameters: $\boldsymbol{\theta_{p^k}} \leftarrow \boldsymbol{\theta_{p^k}} - \gamma_{\boldsymbol{p}} \nabla_{\boldsymbol{\theta_{p^k}}} (\mathcal{L}_{MSE}^k + \mathcal{L}_{MI}^k)$ (Eq. 18).
15:         **end for**
16:     **end for**
17:     Let $\{\tilde{\boldsymbol{p}}^k\}_{k=1}^{K}$ be the prompts from updated $\boldsymbol{\theta_{p^k}}$.
18:     **Step 3: Upper-level Optimization (Primary Model Training)**
19:     **if** $A_s \leq t \leq A_e$ **then**
20:         Initialize best candidate $\{\tilde{\boldsymbol{p}}_{\text{best}}^k\} \leftarrow \{\tilde{\boldsymbol{p}}^k\}$.
21:         Compute $\mathcal{L}_{MSE}^{\text{best}} = \mathcal{L}_{MSE}$ and Grad-Sim$_{\text{best}}$ = Grad-Sim.
22:         **for** $s = 1$ to $n_s$ **do**            ▷ Iterative search
23:            Perturb prompts: $\{(\tilde{\boldsymbol{p}}^k)'\} \leftarrow \{\tilde{\boldsymbol{p}}^k\} + \mathcal{U}(-\rho, \rho)$ (Eq. 19).
24:            Compute $\mathcal{L}_{MSE}^k$ and Grad-Sim.
25:            **if** Grad-Sim > Grad-Sim$_{\text{best}}$ **and** $\mathcal{L}_{MSE} < \mathcal{L}_{\text{best}}$ **then**
26:                Update best candidate $\{\tilde{\boldsymbol{p}}_{\text{best}}^k\} \leftarrow \{(\tilde{\boldsymbol{p}}^k)'\}$.
27:                $\mathcal{L}_{\text{best}} \leftarrow \mathcal{L}_{MSE}$, Grad-Sim$_{\text{best}} \leftarrow$ Grad-Sim.
28:            **end if**
29:         **end for**
30:         $\tilde{\boldsymbol{p}}_{\text{final}} \leftarrow \sum_{k=1}^{K} \tilde{\boldsymbol{p}}_{\text{best}}^k$.
31:     **else**
32:         $\tilde{\boldsymbol{p}}_{\text{final}} \leftarrow \sum_{k=1}^{K} \tilde{\boldsymbol{p}}^k$.
33:     **end if**
34:     Compute final loss $\mathcal{L}_{MSE}(\boldsymbol{C}^{1:T_{obs}}, \tilde{\boldsymbol{p}}_{\text{final}})$.
35:     Update primary model: $\boldsymbol{\theta_c} \leftarrow \boldsymbol{\theta_c} - \gamma_{\boldsymbol{c}} \nabla_{\boldsymbol{\theta_c}} \mathcal{L}_{MSE}$ (Eq. 21).
36:     **Step 4: Trajectory Simulation for Data Augmentation**
37:     Generate unseen trajectories and select by a vision language model for augmentation.
38: **end for**
39: **return** $\boldsymbol{\theta_c}, \{\boldsymbol{\theta_{p^k}}\}_{k=1}^{K}$.

---

# E   MORE EXPERIMENT RESULTS

## E.1   DATASET DETAILS

We evaluate our method on four challenging simulation datasets. These datasets span two categories: physical dynamics and molecular dynamics. The crucial distinction between in-distribution (ID) and out-of-distribution (OOD) test sets is achieved by sampling system parameters from disjoint value ranges.

**Physical Dynamics: Springs & Charged.** The first two benchmarks model the dynamics of 10 interacting particles in a 2D space. **Springs** dataset features a system where particle interactions are governed by Hooke's Law, simulating an interconnected network of springs. The **Charged** dataset is similar in structure but models particles interacting via electrostatic forces, where attraction and repulsion occur with equal probability. In both systems, the interaction topology between particles remains fixed. The OOD challenge is created by varying parameters such as the box size, initial velocity, and interaction strength.

**Molecular Dynamics: 5AWL & 2N5C.** To assess performance on more complex, high-dimensional systems, we employ two molecular dynamics datasets, **5AWL** and **2N5C**. These were constructed based on simulations of two proteins obtained from the RCSB database. The trajectories are generated using Langevin Dynamics (García-Palacios & Lázaro, 1998) under an NPT (isothermal-isobaric) ensemble, capturing the motion of atoms within a solvent environment. For these datasets, the domain shift is induced by varying the simulation's temperature, pressure, and frictional coefficient.

### E.2 BASELINES FOR COMPARISON

To comprehensively evaluate the performance of our proposed method, we selected a diverse set of representative baselines, spanning multiple categories from classical sequential models to state-of-the-art dynamic graph models.

**LSTM** ((Hochreiter & Schmidhuber, 1997)): As a foundational model for sequence prediction, Long Short-Term Memory (LSTM) is distinguished by its sophisticated gating mechanism, which enables it to effectively capture long-term dependencies within data sequences, surpassing classic RNNs.

**GRU** ((Cho et al., 2014)): The Gated Recurrent Unit (GRU) is another popular recurrent architecture that employs a simplified gating structure to manage information flow. It often delivers performance comparable to LSTM but with greater computational efficiency.

**NODE** ((Chen et al., 2018)): Neural Ordinary Differential Equations (NODEs) pioneered the concept of continuous-depth neural networks by framing the architecture as an ordinary differential equation (ODE) solver. The method has demonstrated strong efficacy in time-series forecasting.

**LG-ODE** ((Huang et al., 2020)): This model integrates Graph Neural Networks (GNNs) with neural ODEs, enabling it to capture continuous interacting dynamics even from irregularly-sampled or partially observed data.

**MP-NODE** ((Chen et al., 2022b)): This method explicitly integrates the message-passing mechanism into the neural ODE framework, aiming to capture the dynamic relationships at the sub-system level within homogeneous systems.

**SocialODE** ((Wen et al., 2022)): Designed specifically for multi-agent trajectory forecasting, SocialODE utilizes a neural ODE architecture to simulate the continuous evolution of agent states and their interactions, achieving remarkable performance.

**HOPE** ((Luo et al., 2023)): A recently proposed graph ODE method, HOPE leverages a twin encoder to learn latent representations, which are then fed into a high-order graph ODE to learn long-term correlations in complex dynamical systems.

**PGODE** ((Luo et al., 2024)): As a strong baseline in our comparison, Prototypical Graph ODE (PGODE) is designed to address out-of-distribution generalization. It enhances model expressivity by employing a weighted combination of multiple GNN "prototypes," where the combination weights are determined by disentangled object-level and system-level contexts.

**Pioneer** ((Sun et al., 2025)): This work presents a physics-informed graph ODE on Riemannian manifolds , which models entropy-increasing dynamic systems by introducing a constrained Ricci flow to ensure adherence to physical laws and the system's intrinsic geometry.

Table 4: Mean Squared Error (MSE) $\times 10^{-2}$ on Charged dataset, where $q$ denotes position and $v$ denotes velocity error.

| Prediction length | 12 (ID) | | 24 (ID) | | 36 (ID) | | 12 (OOD) | | 24 (OOD) | | 36 (OOD) | |
| Methods | $q$ | $v$ | $q$ | $v$ | $q$ | $v$ | $q$ | $v$ | $q$ | $v$ | $q$ | $v$ |
|---|---|---|---|---|---|---|---|---|---|---|---|---|
| LSTM | 0.795 | 3.029 | 2.925 | 3.734 | 6.569 | 4.331 | 1.127 | 3.027 | 3.988 | 3.640 | 8.185 | 4.221 |
| GRU | 0.781 | 2.997 | 2.805 | 3.640 | 5.969 | 4.147 | 1.042 | 3.028 | 3.747 | 3.636 | 7.515 | 4.101 |
| NODE | 0.776 | 2.770 | 3.014 | 3.441 | 6.668 | 4.043 | 1.124 | 2.844 | 3.931 | 3.563 | 8.497 | 4.737 |
| LG-ODE | 0.759 | 2.368 | 2.526 | 3.314 | 5.985 | 5.618 | 0.932 | 2.551 | 3.018 | 3.589 | 6.795 | 6.365 |
| MP-NODE | 0.740 | 2.455 | 2.458 | 3.664 | 5.625 | 6.259 | 0.994 | 2.555 | 2.898 | 3.835 | 6.084 | 6.797 |
| SocialODE | 0.662 | 2.335 | 2.441 | 3.252 | 6.410 | 4.912 | 0.894 | 2.420 | 2.894 | 3.402 | 6.292 | 6.340 |
| HOPE | 0.614 | 2.316 | 3.076 | 3.381 | 8.567 | 8.458 | 0.878 | 2.475 | 3.685 | 3.430 | 10.953 | 9.120 |
| Pioneer | 0.524 | 1.978 | 2.627 | 2.888 | 5.319 | 4.226 | 0.750 | 2.114 | 3.148 | 2.930 | 7.357 | 3.792 |
| PGODE | 0.578 | 2.196 | 2.037 | 2.648 | 4.804 | 3.551 | 0.802 | 2.135 | 2.584 | 2.663 | 5.703 | 3.703 |
| **PEACE** (ours) | **0.547** | **1.619** | **1.893** | **1.814** | **4.269** | **2.027** | **0.663** | **1.824** | **2.337** | **2.253** | **5.453** | **2.174** |

### E.3 IMPLEMENTATION DETAILS

In our experiments, the system parameters for training, validation, and in-distribution (ID) test samples are randomly drawn from the training parameter space $\Omega_{\text{train}}$, whereas out-of-distribution (OOD) samples are randomly generated from $\Omega_{\text{OOD}}$, introducing shifts compared to the training distribution. The conditional length is fixed to 12, and we evaluate three different prediction horizons: 12, 24, and 36.

All methods, including our proposed PEACE, are implemented in Python 3.10 using PyTorch 2.7.0 (Paszke et al., 2017) and the torchdiffeq package (Kidger et al., 2021), and all experiments are conducted on four NVIDIA 3090 GPUs. The ODE solver is the fourth-order Runge-Kutta method provided by torchdiffeq. The number of prompts is set to 3 by default. We adopt the AdamW optimizer (Loshchilov & Hutter, 2017) with initial learning rates of $5 \times 10^{-5}$ for the upper layers and $1 \times 10^{-3}$ for the lower layers. The EMA coefficient is set to 0.9. The batch size is set to 16 for both physical and molecular dynamics datasets.

During training, we employ a combined gradient update strategy that leverages both global and prompt-specific gradients to ensure gradient consistency and improve model generalization. Specifically, the global gradient updates the backbone parameters, while each prompt is updated only with its own gradient. Alignment with the global gradient is performed to mitigate gradient conflicts.

We use mean squared error (MSE) as the evaluation metric. To ensure reproducibility, the random seed is set to 42. Additionally, an early stopping strategy is applied: training is terminated if the loss does not decrease for 10 consecutive epochs.

### E.4 PERFORMANCE COMPARISON

The detailed results on the Charged and 2N5C datasets are reported in Tables 4 and 5, respectively To further validate the effectiveness of PEACE, we provide additional baseline comparisons in the appendix, including AgentFormer (Yuan et al., 2021b), NRI (Kipf et al., 2018), and I-GPODE (Yıldız et al., 2022), as shown in Table 6. We also include two representative equivariance-based methods, EGNN (Satorras et al., 2021) and EqMotion (Xu et al., 2023), with results presented in Table 7. In both of these experiment groups, PEACEconsistently achieves the best performance under both ID and OOD settings. In the more fine-grained evaluation of position and velocity along the x/y directions ($q_x/q_y/v_x/v_y$, Tables 8 and 9), our method may be slightly suboptimal in individual directions; however, when averaging over both directions, it still maintains the best overall performance. This demonstrates that our model exhibits strong robustness in global prediction, where occasional fluctuations in a single coordinate direction do not undermine its overall superiority over existing methods.

### E.5 ABLATION STUDY

We further performed ablation experiments on the 2N5C dataset to provide a more comprehensive analysis. As summarized in Table 10, the results show that the full version of PEACE consistently outperforms all model variants across different settings, thereby confirming the effectiveness of each individual component.

Table 5: Mean Squared Error (MSE) $\times 10^{-2}$ on 2N5C dataset.

| Prediction length Methods | 12 (ID) | | | 24 (ID) | | | 12 (OOD) | | | 24 (OOD) | | |
|---|---|---|---|---|---|---|---|---|---|---|---|---|
| | $q_x$ | $q_y$ | $q_z$ | $q_x$ | $q_y$ | $q_z$ | $q_x$ | $q_y$ | $q_z$ | $q_x$ | $q_y$ | $q_z$ |
| LSTM | 0.260 | 0.226 | 0.397 | 0.338 | 0.295 | 0.429 | 0.328 | 0.221 | 0.524 | 0.383 | 0.287 | 0.507 |
| GRU | 0.284 | 0.296 | 0.349 | 0.334 | 0.339 | 0.363 | 0.351 | 0.368 | 0.379 | 0.403 | 0.393 | 0.374 |
| NODE | 0.221 | 0.210 | 0.260 | 0.307 | 0.284 | 0.328 | 0.291 | 0.264 | 0.279 | 0.366 | 0.347 | 0.387 |
| LG-ODE | 0.217 | 0.188 | 0.192 | 0.282 | 0.241 | 0.268 | 0.264 | 0.228 | 0.232 | 0.365 | 0.312 | 0.340 |
| MP-NODE | 0.185 | 0.192 | 0.223 | 0.283 | 0.280 | 0.341 | 0.230 | 0.255 | 0.237 | 0.324 | 0.353 | 0.322 |
| SocialODE | 0.196 | 0.171 | 0.181 | 0.257 | 0.228 | 0.241 | 0.234 | 0.213 | 0.216 | 0.338 | 0.299 | 0.305 |
| HOPE | 0.184 | 0.191 | 0.222 | 0.265 | 0.278 | 0.347 | 0.256 | 0.251 | 0.273 | 0.334 | 0.330 | 0.350 |
| Pioneer | 0.157 | 0.163 | 0.189 | 0.226 | 0.237 | 0.296 | 0.218 | 0.214 | 0.233 | 0.285 | 0.281 | 0.299 |
| PGODE | 0.148 | 0.142 | 0.157 | 0.196 | 0.202 | 0.211 | 0.168 | 0.180 | 0.191 | 0.246 | 0.273 | 0.272 |
| **PEACE** (ours) | **0.133** | **0.119** | **0.136** | **0.138** | **0.117** | **0.176** | **0.120** | **0.135** | **0.155** | **0.199** | **0.187** | **0.207** |

Table 6: Mean Squared Error (MSE) $\times 10^{-2}$ of NRI, AgentFormer and I-GPODE on dynamics simulations.

| Dataset | Prediction length Methods | 12 (ID) | | 24 (ID) | | 36 (ID) | | 12 (OOD) | | 24 (OOD) | | 36 (OOD) | |
|---|---|---|---|---|---|---|---|---|---|---|---|---|---|
| | | $q$ | $v$ | $q$ | $v$ | $q$ | $v$ | $q$ | $v$ | $q$ | $v$ | $q$ | $v$ |
| *Springs* | NRI | 0.103 | 0.425 | 0.210 | 0.681 | 0.693 | 2.263 | 0.119 | 0.472 | 0.246 | 0.770 | 0.807 | 2.406 |
| | AgentFormer | 0.115 | 0.163 | 0.0.202 | 0.517 | 1.656 | 1.691 | 0.157 | 0.195 | 0.243 | 0.505 | 1.875 | 1.913 |
| | I-GPODE | 0.159 | 0.479 | 0.746 | 3.002 | 1.701 | 7.433 | 0.173 | 0.498 | 0.796 | 3.193 | 1.818 | 7.322 |
| | **PEACE** (ours) | **0.018** | **0.119** | **0.031** | **0.107** | **0.279** | **0.704** | **0.040** | **0.136** | **0.086** | **0.172** | **0.224** | **0.753** |
| *Charged* | NRI | 0.901 | 2.702 | 3.225 | 3.346 | 7.770 | 4.543 | 1.303 | 2.726 | 3.678 | 3.548 | 8.055 | 4.752 |
| | AgentFormer | 1.076 | 2.476 | 3.631 | 3.044 | 7.513 | 3.944 | 1.384 | 2.514 | 4.224 | 3.199 | 8.985 | 4.002 |
| | I-GPODE | 1.044 | 2.818 | 3.407 | 3.751 | 7.292 | 4.570 | 1.322 | 2.715 | 3.805 | 3.521 | 8.011 | 4.056 |
| | **PEACE** (ours) | **0.547** | **1.619** | **1.893** | **1.814** | **4.269** | **2.027** | **0.663** | **1.824** | **2.337** | **2.253** | **5.453** | **2.174** |

Table 7: Performance comparison with EGNN, EqMotion, and PGODE on physical dynamics simulations (MSE $\times 10^{-2}$).

| Dataset | Springs | | | | Charged | | | |
|---|---|---|---|---|---|---|---|---|
| Prediction length | 12 (ID) | | 12 (OOD) | | 12 (ID) | | 12 (OOD) | |
| Methods | $q_x$ | $q_y$ | $q_x$ | $q_y$ | $q_x$ | $q_y$ | $q_x$ | $q_y$ |
| EGNN | 0.140 | 0.147 | 0.150 | 0.149 | 2.092 | 2.227 | 2.139 | 2.244 |
| EqMotion | 0.077 | 0.080 | 0.084 | 0.080 | 0.807 | 0.893 | 0.867 | 0.936 |
| **PEACE** (ours) | **0.017** | **0.019** | **0.048** | **0.032** | **0.346** | **0.748** | **0.517** | **0.809** |

Table 8: Fine-grained Mean Squared Error (MSE) $\times 10^{-2}$ on Springs dataset.

| Dataset | Prediction length Methods | 12 | | | | 24 | | | | 36 | | | |
|---|---|---|---|---|---|---|---|---|---|---|---|---|---|
| | | $q_x$ | $q_y$ | $v_x$ | $v_y$ | $q_x$ | $q_y$ | $v_x$ | $v_y$ | $q_x$ | $q_y$ | $v_x$ | $v_y$ |
| ID | LSTM | 0.324 | 0.250 | 0.909 | 0.931 | 0.679 | 0.638 | 2.695 | 2.623 | 1.253 | 1.304 | 5.023 | 6.434 |
| | GRU | 0.496 | 0.291 | 0.565 | 0.628 | 0.873 | 0.623 | 1.711 | 2.001 | 1.368 | 1.128 | 2.980 | 3.912 |
| | NODE | 0.165 | 0.148 | 0.649 | 0.479 | 0.722 | 0.621 | 2.534 | 2.293 | 1.683 | 1.534 | 6.323 | 6.142 |
| | LG-ODE | 0.077 | 0.077 | 0.264 | 0.272 | 0.174 | 0.135 | 0.449 | 0.576 | 0.613 | 0.441 | 1.757 | 2.528 |
| | MP-NODE | 0.080 | 0.072 | 0.222 | 0.263 | 0.237 | 0.105 | 0.407 | 0.506 | 0.866 | 0.335 | 1.469 | 2.006 |
| | SocialODE | 0.069 | 0.068 | 0.205 | 0.315 | 0.138 | 0.120 | 0.391 | 0.630 | 0.429 | 0.400 | 1.751 | 2.624 |
| | HOPE | 0.087 | 0.053 | 0.152 | 0.200 | 0.571 | 0.342 | 0.707 | 1.206 | 2.775 | 2.175 | 4.412 | 6.405 |
| | Pioneer | 0.074 | 0.045 | 0.130 | 0.171 | 0.488 | 0.292 | 0.604 | 1.030 | 2.371 | 1.858 | 3.769 | 5.472 |
| | PGODE | 0.033 | 0.037 | **0.122** | 0.127 | 0.074 | 0.066 | 0.239 | 0.286 | 0.318 | 0.273 | 1.186 | 1.466 |
| | **PEACE** (ours) | **0.017** | **0.019** | 0.125 | **0.116** | **0.032** | **0.031** | 0.224 | 0.173 | **0.272** | **0.173** | **0.764** | **0.644** |
| OOD | LSTM | 0.499 | 0.449 | 1.086 | 1.227 | 1.019 | 0.857 | 2.847 | 2.466 | 1.768 | 1.415 | 5.154 | 5.293 |
| | GRU | 0.714 | 0.469 | 0.713 | 0.703 | 1.280 | 0.905 | 1.795 | 2.096 | 1.844 | 1.497 | 2.852 | 3.994 |
| | NODE | 0.246 | 0.209 | 0.997 | 0.585 | 0.876 | 0.687 | 2.790 | 2.269 | 2.002 | 1.663 | 6.349 | 5.670 |
| | LG-ODE | 0.093 | 0.083 | 0.272 | 0.327 | 0.185 | 0.172 | 0.463 | 0.661 | 0.684 | 0.545 | 1.767 | 2.645 |
| | MP-NODE | 0.107 | 0.081 | 0.230 | 0.268 | 0.299 | 0.126 | 0.420 | 0.528 | 0.967 | 0.386 | 1.464 | 1.969 |
| | SocialODE | 0.082 | 0.076 | 0.221 | 0.350 | 0.151 | 0.156 | 0.414 | 0.726 | 0.488 | 0.495 | 1.793 | 2.826 |
| | HOPE | 0.094 | 0.058 | 0.178 | 0.264 | 0.506 | 0.523 | 1.031 | 1.603 | 2.369 | 2.251 | 3.701 | 8.291 |
| | Pioneer | 0.080 | 0.050 | 0.152 | 0.226 | 0.432 | 0.447 | 0.881 | 1.370 | 2.024 | 1.923 | 3.162 | 7.083 |
| | PGODE | **0.046** | 0.048 | 0.133 | 0.144 | 0.094 | 0.081 | 0.286 | 0.297 | 0.336 | 0.281 | 1.360 | 1.313 |
| | **PEACE** (ours) | 0.048 | **0.032** | **0.130** | 0.142 | **0.093** | **0.079** | **0.186** | **0.230** | 0.326 | **0.232** | 0.814 | 0.692 |

Table 9: Fine-grained Mean Squared Error (MSE) $\times 10^{-2}$ on Charged dataset.

| Dataset | Prediction length Methods | 12 | | | | 24 | | | | 36 | | | |
|---|---|---|---|---|---|---|---|---|---|---|---|---|---|
| | | $q_x$ | $q_y$ | $v_x$ | $v_y$ | $q_x$ | $q_y$ | $v_x$ | $v_y$ | $q_x$ | $q_y$ | $v_x$ | $v_y$ |
| ID | LSTM | 0.743 | 0.846 | 2.913 | 3.145 | 2.797 | 3.052 | 3.605 | 3.863 | 6.477 | 6.660 | 4.240 | 4.423 |
| | GRU | 0.764 | 0.799 | 2.931 | 3.063 | 2.709 | 2.901 | 3.572 | 3.709 | 5.657 | 6.281 | 4.068 | 4.227 |
| | NODE | 0.743 | 0.808 | 2.764 | 2.777 | 2.913 | 3.114 | 3.432 | 3.451 | 6.468 | 6.868 | 3.997 | 4.089 |
| | LG-ODE | 0.736 | 0.783 | 2.322 | 2.414 | 2.320 | 2.731 | 3.361 | 3.268 | 5.188 | 6.782 | 6.194 | 5.043 |
| | MP-NODE | 0.720 | 0.759 | 2.414 | 2.496 | 2.379 | 2.536 | 3.589 | 3.738 | 5.636 | 5.614 | 5.472 | 7.046 |
| | SocialODE | 0.630 | 0.695 | 2.311 | 2.358 | 2.252 | 2.631 | 3.509 | 2.995 | 5.743 | 7.076 | 5.701 | 4.122 |
| | HOPE | 0.593 | 0.635 | 2.295 | 2.337 | 3.214 | 2.938 | 3.279 | 3.482 | 9.289 | 7.845 | 8.406 | 8.511 |
| | Pioneer | 0.507 | 0.633 | 1.961 | 1.990 | 2.746 | 2.510 | 2.801 | 2.975 | 7.936 | 6.702 | 7.182 | 7.271 |
| | PGODE | 0.555 | **0.600** | 2.164 | 2.228 | 1.940 | 2.134 | 2.624 | 2.673 | 4.449 | 5.159 | 3.778 | 3.324 |
| | **PEACE (ours)** | **0.346** | 0.748 | **0.110** | **0.128** | **1.902** | **1.884** | **1.980** | **1.647** | **3.632** | **4.907** | **1.534** | **2.347** |
| OOD | LSTM | 1.130 | 1.123 | 3.062 | 2.992 | 4.026 | 3.950 | 3.768 | 3.512 | 7.934 | 8.435 | 4.517 | 3.925 |
| | GRU | 1.072 | 1.012 | 3.108 | 2.948 | 3.893 | 3.602 | 3.844 | 3.428 | 6.970 | 8.061 | 4.485 | 3.718 |
| | NODE | 1.185 | 1.062 | 2.956 | 2.732 | 4.057 | 3.804 | 3.645 | 3.480 | 8.622 | 8.372 | 5.097 | 4.376 |
| | LG-ODE | 0.999 | 0.866 | 2.581 | 2.521 | 2.797 | 3.239 | 4.200 | 2.978 | 5.996 | 7.593 | 8.422 | 4.309 |
| | MP-NODE | 1.092 | 0.897 | 2.487 | 2.623 | 2.967 | 2.828 | 3.670 | 4.001 | 6.051 | 6.118 | 6.029 | 7.566 |
| | SocialODE | 0.865 | 0.924 | 2.481 | 2.359 | 2.610 | 3.177 | 3.968 | 2.836 | 5.482 | 7.102 | 8.530 | 4.150 |
| | HOPE | 0.839 | 0.918 | 2.466 | 2.484 | 3.586 | 3.783 | 3.417 | 3.442 | 11.254 | 10.652 | 10.133 | 8.107 |
| | Pioneer | 0.717 | 0.884 | 2.107 | 2.122 | 3.063 | 3.232 | 2.919 | 2.940 | 9.615 | 9.100 | 8.657 | 6.926 |
| | PGODE | 0.739 | 0.865 | 2.159 | 2.110 | 2.524 | 2.643 | 2.704 | 2.623 | 5.748 | 5.659 | 4.017 | 3.389 |
| | **PEACE ( ours)** | **0.517** | **0.809** | **1.537** | **1.701** | **2.441** | **2.233** | **2.163** | **2.343** | **5.653** | **5.283** | **2.311** | **2.037** |

Table 10: Ablation study on *2N5C* with a prediction length of 24.

| Dataset Variable | *2N5C* (ID) | | | *2N5C* (OOD) | | |
|---|---|---|---|---|---|---|
| | $q_x$ | $q_y$ | $q_z$ | $q_x$ | $q_y$ | $q_z$ |
| W/O PROXY MODELS | 0.272 | 0.228 | 0.342 | 0.384 | 0.347 | 0.386 |
| W/O BI-LEVEL OPTIMIZATION | 0.198 | 0.165 | 0.285 | 0.275 | 0.247 | 0.327 |
| W/O SIMULATION | 0.184 | 0.159 | 0.260 | 0.254 | 0.238 | 0.296 |
| W/O VLM | 0.172 | 0.144 | 0.244 | 0.239 | 0.224 | 0.259 |
| **PEACE (ours)** | **0.138** | **0.117** | **0.176** | **0.199** | **0.187** | **0.207** |

### E.6 ADDITIONAL VISUALIZATION.

Figure 6 follow the same setup as in the main text, presenting comparisons among PEACE, HOPE, PGODE, and the ground truth. Consistent with the observations reported in the main paper, PEACE produces trajectories that remain close to the ground truth, even in scenarios with complex particle interactions or large numbers of atoms. In contrast, HOPE and PGODE exhibit noticeable deviations and instability. These supplementary visualizations further corroborate the stability, robustness, and generalization ability of PEACE across both physical and molecular systems.

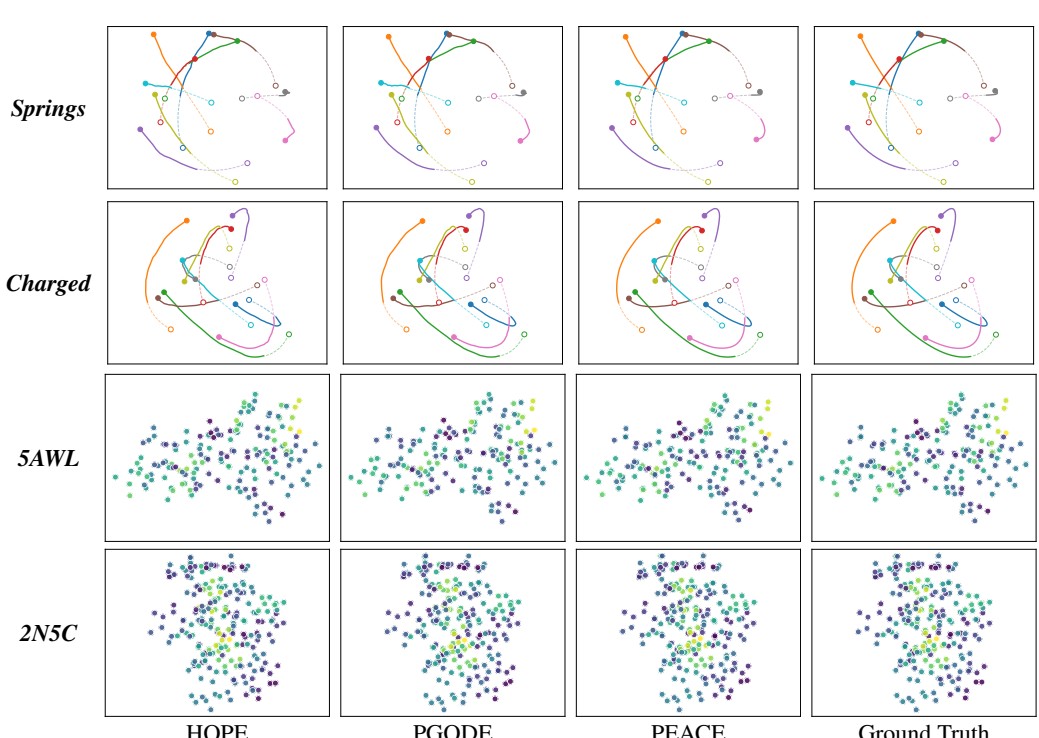

Figure 6: Visualization of different methods on *Springs* and *Charged*. Dashed lines denote observed trajectories, solid lines represent predicted trajectories with a length of 12 steps, open circles indicate starting points, and filled circles mark ending points.

