# OpenReview forum: "Prototypical Environment-aware Proxy with Coordinated Optimization for Multi-agent Dynamics Modeling"
_ICLR.cc/2026/Conference — ICLR 2026 Conference Withdrawn Submission_

### Official Review · Reviewer_UXj9 · 2025-10-29

**Soundness:** 2
**Presentation:** 2
**Contribution:** 1
**Rating:** 2
**Confidence:** 3

**Summary:**

The paper introduces PEACE, a framework for modeling multi-agent dynamics under distribution shifts. It combines temporal graph neural networks with environment-specific proxy prompts and a bi-level optimization scheme to align learning across environments. A vision-language model filters synthetic trajectories for data augmentation. Experiments on physical and molecular dynamics datasets show improved robustness and accuracy over prior graph ODE methods like HOPE and PGODE.

**Strengths:**

The paper tackles an important and timely problem—improving out-of-distribution generalization in multi-agent dynamics modeling. It integrates known ideas such as graph ODEs, prompt-based proxies, and coordinated optimization into a unified framework that performs strongly across several datasets. While not conceptually groundbreaking, the approach is well engineered, with solid experimental coverage and clear empirical gains over prior baselines. The paper is clearly written, the figures aid understanding, and the methodology could influence future work on robust simulation and control under varying environments.

**Weaknesses:**

1.    The paper builds on prior graph ODE and multi-agent dynamics work (e.g., HOPE, PGODE). The idea of using environment-specific prototypes or proxies is not new; PEACE mainly reuses existing elements under different names.

2.    The model combines GNNs, ODEs, prompts, and bi-level optimization without a clear rationale. Theorem 3.1 is generic and does not explain why the proposed coordination improves learning or generalization.

3.    The use of a vision-language model for trajectory filtering feels ad-hoc. The paper does not explain why this step is necessary or how it compares to simpler validation methods.

4.   Results show better MSEs, but the analysis is shallow. There is no evidence of how gradient alignment or prompt diversity actually help. Ablations only test component removal, not mechanism.

5.    Too many modules are added without a unifying idea. The paper would benefit from a simpler formulation and clearer explanation of what each part contributes.

**Questions:**

1.    How does PEACE differ fundamentally from PGODE or HOPE beyond the use of “proxy prompts”? Is there a principled reason why the proposed proxy mechanism should generalize better under environment shifts? A clear conceptual distinction would help clarify the contribution.

2.    Can the authors provide a stronger theoretical argument or empirical evidence linking gradient coordination to improved OOD robustness? For instance, is there measurable gradient alignment across environments or a stability analysis supporting the design?

3.    Why is a VLM necessary for filtering generated trajectories? Could simpler uncertainty-based or energy-based criteria achieve the same effect? Quantitative ablation on this step would clarify its real benefit.

4.    Can the authors visualize or analyze what each proxy model (prompt) learns? For example, do different proxies correspond to distinct physical regimes or parameter ranges? This could validate the intuition behind “environment-aware” modeling.

5.   Have the authors tested PEACE on systems that differ structurally (e.g., fluid vs. particle dynamics) to show true cross-domain generalization? Current datasets are closely related and may not reflect real distribution shifts.

6.    Is every module essential? Could similar performance be achieved with a simpler architecture, e.g., a single graph ODE with regularization? A discussion on the trade-off between complexity and gain would help.

---

### Official Review · Reviewer_6UH5 · 2025-10-30

**Soundness:** 2
**Presentation:** 3
**Contribution:** 3
**Rating:** 4
**Confidence:** 3

**Summary:**

Multi-agent dynamic systems are widely used in fields such as physical simulation and molecular dynamics. However, existing modeling methods still face several key challenges: training and test data often violate the independent and identically distributed (i.i.d.) assumption, models are sensitive to distribution shifts, and the training process is prone to instability. To address these issues, this paper proposes a modeling framework based on the collaboration between a primary model and multiple proxy models. In this framework, the primary model learns general dynamic patterns of the system using temporal graph structures, while the multiple proxy models dynamically adapt to different environmental changes via Graph Ordinary Differential Equations (ODEs). These components are jointly trained through a coordinated optimization mechanism, and a Vision-Language Model (VLM) is introduced to enhance data quality. Empirical findings confirm the substantial superiority of the proposed approach in terms of both forecasting precision and model generalization.

**Strengths:**

Excellent Generalization Performance: The method simultaneously adapts to distribution shifts caused by temporal evolution and parameter variations, achieving significantly higher prediction accuracy under both in-distribution (ID) and out-of-distribution (OOD) settings compared to state-of-the-art methods.

Synergistic and Efficient Component Design: The modules within the framework have clear division of labor and complementary functions—prototypical prompts capture environment-specific characteristics, the VLM filters high-quality augmented data, and bi-level optimization ensures gradient consistency, collectively enhancing overall modeling performance.

Theoretical Rigor and Experimental Thoroughness: A strict theoretical derivation provides an error bound for the approximation of the dynamics matrix, while comprehensive performance comparisons, ablation studies, and parameter sensitivity analyses on four types of real-world system datasets thoroughly validate the method's effectiveness.

Broad Applicability: The framework can be transferred to multi-agent systems in various domains, such as physical simulation and molecular dynamics, and demonstrates strong modeling robustness against irregularly sampled and partially observed data.

**Weaknesses:**

Lack of Complexity Analysis: No theoretical or experimental evaluation of the time complexity for the overall algorithm or key components is provided. It is recommended to supplement such an analysis.

Parameter Sensitivity Requires Further Optimization: The performance gain from increasing the number of prompts exhibits saturation beyond a certain threshold. The conditioning length requires manual pre-setting (fixed at 12 in the experiments), and an adaptive parameter adjustment mechanism has yet to be incorporated.

Dependence on External VLM Reliability: The quality of trajectory augmentation relies on the VLM's semantic understanding of physical or molecular dynamics. Incorrect judgments by the VLM may introduce noisy data, potentially compromising training stability.

Generalization in Extreme Scenarios Remains Unverified: The method's adaptability under extremely long prediction horizons or severe parameter shifts (e.g., system parameters far outside the training distribution) has not been experimentally evaluated.

**Questions:**

See Weaknesses.

---

### Official Review · Reviewer_9X5Q · 2025-11-04

**Soundness:** 2
**Presentation:** 2
**Contribution:** 2
**Rating:** 2
**Confidence:** 3

**Summary:**

This paper presents a method called PEACE for modeling multi-agent dynamic systems (e.g., particles interacting with each other, and we want to predict their motion). The authors claim that previous graph neural ODE methods are difficult to generalize to OOD scenes. Therefore, the authors made 3 innovations: (1) PEACE learns multiple prototypical proxy models that simulate different environment behaviors while keeping a primary dynamics model fixed; (2) PEACE has a bi-level optimization in which at one stage only the proxy model’s parameters are updated while at another stage the primary model’s parameters are adjusted to align with the proxy models; (3) a VLM to select plausible synthetic trajectories from the proxies for data augmentation. The authors performed experiments on particle and molecular motion datasets and observed improved prediction accuracy in both ID and OOD scenes.

**Strengths:**

-	Using a proxy model and a bi-level optimization scheme to improve OOD performance is novel.
-	The authors perform lots of experiments against 9 baselines.

**Weaknesses:**

-	Definition of ID v.s. OOD: The distinction between ID and OOD data formulation is critical for readers to understand the importance of the work. In Section E.1, the authors briefly discuss that OOD is achieved by changing some system parameters (e.g., interaction strength). Adding more details here is needed. Furthermore, when forming OOD, changes to each system parameter should be better done individually (i.e., you do not change box size and Hook coefficient at the same time). This way we can more accurately analyze how OOD affects the modeling.
-	The proposed method involves many components (multiple proxy `prompt` models, VLM, bi-level optimization), making it complex to be understood and reproduced.
-	Calling the P in Eq.8 as prompts is misleading. They are not the VLM prompts, but rather like a hidden state. The use of the VLM, though improving the performance, seems very irrelevant to the proxy models, and adding it distracts readers. The VLM introduced a lot of variables that need their own ablation studies.
-	In Table 3 last row, the (q, v) error on Springs (ID) Length 24 is (0.031, 0.107), but in Table 1, the corresponding value is (0.031, **0.199**). Similarly, the Springs (OOD) is (0.086, 0.172), but the same item from Table 1 has value (0.086, **0.208**).
-	Line 407: just observing the improved metrics is not sufficient to conclude that PEACE works better due to the three techniques. This conclusion is more appropriate for the ablation studies.
-	The 4 figures in Fig.3 look very similar. Please explain what the authors observe.
-	The 4 figures in Fig.4 also look very similar, and eyeballing the difference is hard. Please highlight the differences. What are “the regions with sharp directional changes”?

**Questions:**

-	Line 81: we see the errors always decrease, so why does the author say “unstable dynamics learning”?
-	Eq.5: how is the correlation weight determined? Is it learned? Also, is A a temporal sequence of two-dimensional matrices? Or is it 3-dimensional?

---

### Official Review · Reviewer_HV7w · 2025-11-04

**Soundness:** 3
**Presentation:** 3
**Contribution:** 3
**Rating:** 6
**Confidence:** 4

**Summary:**

The paper proposes PEACE (Prototypical Environment-Aware Proxy with Coordinated Optimization) for multi-agent dynamics on temporal graphs under two OOD regimes: (i) temporal shift (long-horizon rollout drift) and (ii) parameter-induced shift (e.g., spring constants, temperature). The method splits learning into:

1. A primary temporal GNN that extracts invariant observation embeddings c;

2. A set of K proxy models realized as time-evolving prototypical prompts pk governed by a Graph-ODE; the fused latent [c,pk] is decoded to forecast futures. Guided trajectory simulation generates candidate rollouts per proxy; a VLM ensemble + self-justification filters plausible ones that are then used as augmentation for that proxy. Finally, training is bi-level: proxies are updated with MSE + MI decoupling (primary frozen), and the primary is updated by maximizing the minimum pairwise gradient cosine similarity (“Grad-Sim”) across proxies, with a brief annealed search over prompt perturbations and parameter-averaging across proxies. A linearized analysis (Theorem 3.1) gives an LTI bound: trajectory error grows with ∥A−A*∥2​, motivating gradient coordination.

Claimed outcome. On physical (Springs/Charged) and molecular systems (e.g., protein trajectories), the approach targets better ID/OOD accuracy at long horizons, attributing gains to (i) proxy prompting, (ii) VLM-filtered augmentation, and (iii) gradient coordination.

**Strengths:**

1. Clear decomposition of invariants vs. environment-specifics. Primary c vs. proxy prompts p + MI decoupling is a principled way to avoid leakage and spurious correlations.

2. Graph-ODE prompts are well-motivated. The state-aware attention in the ODE drift couples' neighbors and adapts across time; fits multi-agent physics intuitions.

3. Bi-level training with explicit gradient-agreement metric. Using min pairwise cosine as a hard-case coordination target is simple and effective; annealed search + parameter averaging ties to ensembling/flat minima intuitions.

4. Theoretical lens. The LTI bound connects coordination to a smaller operator gap ∥A−A*∥, aligning with the method’s goal (even if assumptions are strong).

5. Operational clarity. The training loop is spelled out (inputs, schedules, updates), aiding implementation.

**Weaknesses:**

1. VLM pipeline under-specified: model names, prompt templates, voting/aggregation rule and acceptance thresholds are not given, hurting reproducibility.

2. No MI-loss ablation: the mutual-information term that enforces c⊥p is described but not experimentally isolated.

3. Grad-Sim choice unexplored: the paper uses min pairwise cosine but provides no comparison to mean/median or variance-based alternatives.

4. Missing compute/sensitivity analysis: no GPU-hours, wall-clock, memory numbers, nor sweeps for inner/outer loop hyperparameters (na​,ns​,ρ).

5. Prompt interpretability lacking: no diagnostics (e.g., clustering/correlation with physical parameters) to show proxies correspond to meaningful environment regimes.

**Questions:**

1. VLM specifics & script: which VLM(s) exactly, full prompt files, preprocessing for visualizations, ensemble aggregation rule, seeds, and the decision threshold used for accepting rollouts?

2. VLM ablation: can you report results replacing the VLM ensemble with (a) a single VLM and (b) a simple physics filter (e.g., energy/speed bounds)?

3. MI ablation: please provide the same ablation rows with “w/o MI” (remove the mutual-information loss) for all datasets/horizons in Table 3.

4. Grad-Sim alternatives: can you run and report small ablations using mean pairwise cosine, median pairwise cosine, and a gradient-variance penalty instead of the current min?

5. Compute & hyperparameter sweep: please report GPU-hours and show MSE + Grad-Sim curves while sweeping na​∈{1,3,5}, ns​∈{0,5,10}, and ρ∈{0.01,0.05,0.1}.

6. Prompt diagnostics: provide a 2D embedding (t-SNE/UMAP) of learned prompts colored by environment parameter (e.g., spring constant / temperature) and the correlation between prompt cluster assignment and parameter values.

---

### Note · Authors · 2026-01-22

I have read and agree with the venue's withdrawal policy on behalf of myself and my co-authors.